# Association of exercise and ADHD symptoms: Analysis within an adult general population sample

Rory Tucker[1]*, Claire Williams[1,2], Phil Reed[1]

1 School of Psychology, Faculty of Medicine, Health and Life Science, Swansea University, Swansea, United Kingdom, 2 Elysium Neurological Services, Elysium Healthcare, The Avalon Centre, Swindon, United Kingdom

* 907983@swansea.ac.uk

## Abstract

Given the limitations associated with existing treatments for Attention Deficit/Hyperactive disorder (ADHD), Physical Activity (PA) has been considered as an adjunct therapeutic option. Previous research has generally found that PA reduces ADHD symptoms in children. However, much less research has explored the same effects in adults, and especially females, with ADHD. This cross-sectional study investigated the relationship between PA and ADHD in adults, and whether any relationship was moderated by proxy diagnostic ADHD group, as well as exploring the roles of motivation and forms of exercise. 268 participants completed an online survey measuring proxy ADHD diagnosis; ADHD symptomatology; PA level; forms of exercise performed, and motivation for exercise. In contrast to previous research performed with child participants (which frequently found significant negative correlations), there was no significant relationship between PA level and total ADHD symptomatology in adults, but there was a significant negative correlation between PA level and inattentive ADHD symptomatology. The strength of relationship between PA level and ADHD symptomatology did not differ based on ADHD proxy diagnostic grouping; PA level based on motivation type; or PA level based on total forms of exercise performed. However, it may be that clear relationships between PA and ADHD symptomatology are not easily identifiable in adult populations when only broad, nonspecific variables/measures are used (e.g., only measuring ADHD symptoms as a continuous total score, rather than considering inattentive/hyperactive symptoms separately). Therefore, greater differentiation between ADHD symptoms and subject characteristics (such as gender) might be required to better establish potential relationships and effects in this area, and better inform any potential PA based treatments.

## 1. Introduction

Attention Deficit/Hyperactive disorder (ADHD) is a neurodevelopmental disorder character-ised by a persistent pattern of inattention and/or hyperactivity, along with poor attention and

**Data Availability Statement:** Data is available within public repository, OSD.io. Link to view data: https://osf.io/4jq3n/?view_only=87a0dc676a2247679d3138baa98d1927.

**Funding:** The authors received no specific funding for this work.

**Competing interests:** The authors have declared that no competing interests exist.

memory, to the point that it interferes with development or functioning [1]. For example, various aspects of memory (e.g., working memory; verbal long-term memory) may be compromised to varying degrees [2, 3], and executive functions (e.g., cognitive flexibility, inhibition, and planning) can also be affected [2, 4], contributing to difficulties in organisation, decision-making, and goal-directed behaviour. Moreover, deficits in social cognition can also lead to challenges in forming and maintaining relationships [5], interpreting social cues [6], and adapting behaviour across social contexts [7].

Prevalence rates vary widely, with reported estimates of between 2.2% [8] and 6–7% for children and adolescents [9], and 2.8% [8] to 6.76% [10] for adults. The aforementioned symptoms of ADHD (e.g., inattention, impulsivity, poor decision making, working memory problems) can lead to lifelong challenges, including education and occupational difficulties and instability [11, 12]; social and interpersonal problems [5, 11]; and difficulty managing finances [11]. ADHD has also been associated with suicidal ideation and behaviour [13], as well as an increased risk of both psychiatric (e.g., depression, anxiety, substance misuse [14] and other comorbidities (e.g., sleep disturbance, obesity) [15]. There are also significant economic and societal costs associated with ADHD [16–18]. In the UK for example, the mean annual cost of social care, National Health Service and educational resources per adolescent with ADHD has been estimated at £5,493, with an estimated annual total UK cost in the region of £670 million in 2010 [16].

Current treatments for ADHD include both pharmacological and non-pharmacological approaches, with most guidelines recommending a mixture of both [19]. However, benefits of pharmacological interventions have been reported to diminish after 14–24 months [20, 21], and many individuals with ADHD may be unresponsive to stimulants or unable to tolerate side effects [22]. Similarly, whilst non-pharmacological interventions may be suitable for a wider population, they have been reported to have both reduced efficacy [23, 24] and cost effectiveness, relative to pharmacological approaches [25, 26].

With the need for alternative treatments (that are applicable to a wider population, with fewer side effects, reduced costs, and greater longevity of effect), Physical Activity (PA) and/or exercise (a subset of PA) have been considered as potential approaches [27–31]. PA has demonstrated positive effects in the general population, such as: improving cognitive functions (e.g., planning, scheduling, working memory, task coordination) [32]; reducing depression and anxiety [33] and improving life expectancy [34, 35]. PA has also been shown to be effective when incorporated into treatment interventions for conditions such as depression, anxiety, stress and neurodegenerative diseases, leading to improved mood, quality of life, and executive function [32, 36–39].

Recent evidence also suggests that PA and exercise may have beneficial effects for ADHD symptomatology, by reducing cognitive deficits (e.g., improved attention, inhibition and speed of information processing), negative physical/motor development, and socio-emotional difficulties [27–31]. In male adults with ADHD (or expressing elevated ADHD symptoms), a 20-minute bout of exercise was found to enhance motivation for cognitive tasks and increase feelings of energy [40], and whole-body vibration has been reported to improve attention [41] and neuropsychological test performance [42]. Similarly, a positive association between PA and reduced ADHD symptoms has been found in male adults with ADHD [43]. It may be that PA has similar modes of action as pharmacological treatments, such as increased production of catecholamines linked with ADHD symptoms [44]. Likewise, it is possible that longer-term effects may, at least in part, be due to long-term neurostructural changes in the brain linked with PA [45–47].

However, despite accumulating evidence that exercise and PA may have beneficial impacts on ADHD, there are several shortcomings attached to existing studies—such as a lack of

research exploring certain demographic groups. One such underrepresented group is adults with ADHD, as highlighted by multiple review papers focusing purely on studies with children [28, 29, 48–51], few containing coverage of both children and adults [30, 31, 44], and none dedicated solely to adult-based samples. Additionally, among the reviews that include both children and adults, the majority focus on children. For example, Den Heijer er al. [30] reviewed 25 studies, but only four [40–43] focused on adult participants. As such, the relationship between PA and ADHD remains relatively unexplored in adults compared to children, indicating a significant gap in understanding.

Another major gap in our understanding, is the lack of research into ADHD in females, which is particularly notable in the context of adult ADHD [28, 30]. For example, all four of the studies focussed on adult participants in the review by Den Heijer et al., focussed exclusively on males [40–43]. There are several reasons underpinning a lack of focus on female adult populations, including the lower diagnostic rate amongst females, specifically, and adults in general [9, 52], leading to a desire to restrict the inclusion criteria to the age and gender groups with higher diagnosis rates [43]. However, recent evidence suggests underdiagnosis of ADHD in both adults [8, 53], and specifically, in adult females [54–56]. Thus, addressing this gap becomes even more pressing considering such findings.

There is also limited research into whether the apparent positive effects of PA on ADHD-associated symptoms vary between individuals with or without a formal diagnosis of ADHD. Additionally, it is unclear whether PA may differentially impact on specific subtypes and ADHD symptom expression, such as inattentive versus hyperactive symptomatology. Current findings are equivocal, with some studies suggesting that PA leads to greater positive effects on ADHD-associated symptoms for individuals with ADHD [41, 57, 58], while others suggest the opposite effect [59]. Mixed findings could be due to studies using a dichotomous classification of ADHD based solely on the presence/absence of a *clinical* diagnosis. This approach might result in undiagnosed individuals with ADHD either being missing from the ADHD group and/or contained within the non-ADHD group, and overlooks how PA may have varying impacts on individuals with specific symptoms of ADHD. Determining whether the relationship between PA and ADHD symptomatology is different between ADHD and non-ADHD diagnosed individuals, alongside exploring under researched demographics such as age and gender, is essential. Such insights are crucial for informing potential PA based treatments and determining which populations may benefit most from the treatment and in what way.

Furthermore, research into potential moderating factors on the effect of PA on ADHD symptoms has received scant attention, including motivation for exercise [60] and form of exercise [29, 61]. Previous research has suggested that motivation can affect participation in exercise [60, 62] (with frequency, intensity, and duration of exercise all positively correlated with a more autonomous sense of motivation) [62] and that the form and combination of exercise can affect ADHD symptoms differently [29, 61].

Given the above, the aim of this study was to investigate the relationship between levels of PA and ADHD symptoms in adults, while exploring the effects of potential moderating variables. To overcome some of the problems noted above relating to adults and female representation, a further aim was to investigate relationships among participants who may not have a formal diagnosis, but who may reach psychometrically-defined subclinical (i.e., proxy) thresholds for ADHD. Based on the previous literature, it was hypothesised that: 1) there would be a significant negative correlation between PA level and ADHD symptomatology; 2) there would be no significant difference between the relationship between PA level and ADHD symptomatology in participants with and without a subclinical, proxy diagnosis of ADHD; 3) there would be a significant difference in PA level based on motivation for exercise; and 4) there would be a significant difference in ADHD symptomatology based on main form of exercise.

## 2. Method

### 2.1. Design

The study was a cross-sectional observational study. The primary variables of interest were PA, ADHD symptomatology, and ADHD proxy diagnosis. Potential moderating variables were demographic variables (e.g., age; gender), mood, hours' sleep, and comorbidities. Further variables of interest were forms of exercise, motivation for exercise, and fitness tracking use.

### 2.2. Participants

Participants were recruited online via social media adverts, email invitations, Survey Circle [63], and a University Psychology participant pool. The study was advertised as: '*Physical exercise*: *Links with attention*, *organisation and behavioural inhibition in adults*', and participants were not offered any financial reimbursement or compensation for taking part. Following a detailed information page, participants provided informed consent via an electronic consent page with Yes/No check boxes. Participants who did not agree to all consent statements were redirected to the end of the survey and their data was not used. To be eligible to take part, participants had to be at least 18 years of age and a resident of the United Kingdom. Further exclusion criteria—applied after data was collected—included: not completing the study in full; having duplicate responses to another respondent with same IP address (likely same participant responding twice); having unusable International Physical Activity Questionnaire–Long Format (IPAQ-L) data (e.g., answers filled in incorrectly; scoring above the maximum limit for the IPAQ-L; having repeated answers for questionnaires (suggesting automatic selection and invalid responses); counting as extreme outliers in the IPAQ-L (defined as exceeding the $3^{rd}$ quartile of IPAQ-L score + 3*IQR), and counting as an outlier in Total Forms of Exercise (TFE) done in the past week (defined as exceeding the $3^{rd}$ quartile of TFE score + 3*IQR). No further inclusion/exclusion criteria were applied.

Participants were recruited between 19/12/2020 and 08/05/2021. Initially, 389 participants accessed the survey, although 121 were subsequently excluded for the following reasons: not providing consent ($n = 13$); being under 18 years old ($n = 1$); not being a resident of the UK ($n = 42$); not completing the study in full ($n = 33$); having duplicate responses to another respondent with same IP address (likely same participant responding twice; $n = 12$); having unusable IPAQ-L data (e.g., answers filled in incorrectly; $n = 6$); scoring above the maximum limit for the IPAQ-L ($n = 7$); having repeated answers for questionnaires ($n = 3$); counting as extreme outliers in the IPAQ-L ($n = 3$), and counting as an outlier in forms of exercise done in the past week ($n = 1$).

This left 268 participants, of whom 200 (74.63%) were female. Age ranged from 18 to 71 (M = 25.88, SD = 9.08) years, with further demographic information presented in supplementary information (S1 Table). Options for the Gender and Ethnicity question were based on the categories recommended for use by the UK government [64].

G-Power calculations indicated that for 95% power, with a rejection criterion of $p < .05$, and medium effect size ($f' = .15$), that 184 participants would be needed for a multiple regression with 12 potential predictors. Any potentially identifiable information (e.g., IP address, location data) was removed during initial data screening and prior to formal analysis. The research was approved by a University Departmental Research Ethics Committee (Reference number 5003).

### 2.3. Materials

The online survey was conducted online using Qualtrics [65]. Participants were asked to supply standard demographic details (age, gender, ethnicity, education, employment) and to

details about their: (1) physical fitness (height, weight, hours of sleep); (2) potential comorbid conditions and medication (e.g., "*Have you ever been diagnosed with any of the following conditions/disorders*" [10 options, such as: none; mood disorder; sleep disorders; other; etc], "*If so do you take any prescribed medication*?"); (3) what types of exercise they performed ("*In the last 7 days which forms of exercise listed here have you done*?" [37 options, such as: none, archery, climbing, football, other, etc]); (4) use of fitness trackers ("*Do you use any fitness trackers*? *If so, what*?"); and (5) motivation for exercise ("*What is your main reason / motivation for exercising*?" [options = don't exercise; for my appearance; self-defence; for social reasons; to lose weight; fitness and health; other]). In addition, the survey contained the following standardised questionnaires (see supplementary material 'S1 Appendix' for their psychometric properties):

**The Adult ADHD Self Report Scale– 6 (ASRS-6) [66, 67].** The ASRS-6 was used to classify the proxy diagnostic status of participants in the current study. It is a 6-item scale listing different symptoms of ADHD and is used as a screening tool for proxy diagnosis levels of ADHD in adults. Respondents are required to use a 5-item Likert scale to indicate the frequency of occurrence of symptoms (0 = *never* to 4 = *very often*). If a respondent has 4 or more scores equal and above 2 for items 1–3, and equal and above scale 3 in items 4–6, then they are considered to have symptoms highly consistent with ADHD (i.e., proxy diagnostic group).

**The ADHD Rating Scale–IV with Adult prompts (ADHD-RS-IV with Adult Prompts) [68].** The ASRS-IV was used to measure Total ADHD Symptomatology (T-ADHD-S), Inattentive ADHD Symptomatology (I-ADHD-S); and Hyperactive ADHD Symptomatology (H-ADHD-S). It is an 18-item self-report questionnaire based on DSM-IV criteria that assesses the severity and frequency of ADHD symptoms in adults (ADHD symptomatology). Each item comprises a collection of prompts (e.g. "*Do you rush through work or activities*?") relating to a specific symptom of ADHD (e.g. carelessness). Respondents respond to each item by how much the prompts affect them on a scale of 0–3 (0 = None; 1 = Mild; 2 = Moderate; 3 = Severe). Each item is given the highest score generated by any of its prompts (e.g. if one prompt generates 3, but the rest 2, the score for the item is 3). Responses are then summed to provide a compositive total score (T-ADHD-S) ranging from 0–54, where higher scores indicate greater severity of ADHD symptoms. The questionnaire can also be broken down into 2 subscales (comprised of 9 items each): inattentive ADHD symptoms (I-ADHD-S), and hyperactive ADHD symptoms (H-ADHD-S). This measure was chosen as it is tailored for adult participants and because of its comprehensive measurement of a range of ADHD symptoms.

**The International Physical Activity Questionnaire–Long Format, last seven days, self-administered (IPAQ-L 7S) [69].** The IPAQ-L was used to measure PA level. It is a 27-item questionnaire used to measure physical activity over the last seven days. Data can be converted to Metabolic Equivalents (MET), defined as the amount of oxygen consumed by the body while sitting at rest. The questionnaire guidelines state the maximum amount of exercise that can be done during a week is 16 hours a day, leading to a total range of 0–53760 METs. The scale can be broken down into four subscales/domains: work; travel; domestic, and leisure. This questionnaire was chosen due to it being well established in related research, the outcome variable being METs (a commonly used outcome variable allowing data to be easily compared across-studies), and its focus on a range of domains.

**Other questionnaires.** To investigate potential moderators, additional variables were examined using various measures such as The Personal Health Questionnaire– 4 [70] (to measure mood); The Big Five Inventory– 10 (to measure personality) [71], and the Body Mass Index (to measure physical fitness). Other self-devised questionnaires were developed to measure average hours slept (e.g., "*How many hours of sleep a day do you get on average*?*)* and fitness app use (e.g., "*Do you use any fitness tracking tools [Fitbit, smartphone apps, etc]*"). However, as these variables were not found to have significant links with key study variables,

they were not included in any further analysis and are not described in detail here (for further detail see supplementary material 'S2 Appendix').

## 2.4. Procedure

Participants accessed the survey online via Qualtrics and consenting participants were asked to complete: demographic questions; the ASRS-6; questions on physical fitness; questions on comorbid conditions and medication; the ADHD-RS-IV with Adult Prompts; the IPAQ-L; questions about what types of exercise they performed, and use of fitness trackers and motivation for exercise. All survey sections required a response (except for the questions in the IPAQ-L a participant would have skipped), ensuring the survey was fully completed. There was no time limit, and after excluding outliers (e.g., exceeding the 3$^{rd}$ quartile of total duration + 3*IQR), average completion time was 14 minutes, 37 seconds. Full procedural details can be found in the supplementary material (S2 Appendix).

## 2.5. Data processing and statistical analysis

All statistical analyses were carried out using IBM SPSS Statistics version 28 and PROCESS v4.0 [72]. Additionally, JASP 0.16.3 was used to generate a correlation matrix.

IPAQ-L data was cleaned in accordance with the guidelines set out by the IPAQ group. Data on comorbidities and forms of exercise were converted to ordinal data, by measuring them as 'Total Number of Comorbidities' (TNC) and 'Total Forms of Exercise' (TFE) respectively. Based on ASRS-6 scores, participants were categorised into either a non-ADHD or proxy diagnostic group (participants who met the threshold of having symptoms highly consistent with ADHD, as outlined in the materials subsection, were counted as ADHD proxy diagnostic participants).

In terms of the statistical approach adopted, none of the variables were normally distributed, so non-parametric analyses were utilised, with median and IQR used as measures of central tendency and dispersion respectively. Correlations between key variables (ADHD symptomatology; PA level; TNC; TFE) were performed using Spearman's Rank correlation. Additionally, when investigating relationships between demographic variables and the two main variables of interest (ADHD symptomatology and PA level), Kruskal-Wallis tests (for education, ethnicity and employment), Mann Whitney U tests (for gender), and Spearman's Rank correlations (for age) were performed. To investigate the relationship between proxy diagnostic group and gender, a chi square test of association was performed.

To investigate whether there was a significant negative correlation between PA level and ADHD symptomatology (hypothesis 1), a three-stage multiple hierarchical regression was conducted. ADHD symptomatology was the variable being predicted and PA level, age and TNC were entered as predictor variables. A series of Spearman's rho correlations were also performed within the different subgroups (gender and proxy diagnostic groups) to further investigate the relationships between PA and ADHD symptom subtypes.

A moderation analysis was then employed to investigate whether the relationship between PA level and ADHD symptomatology differed in participants with or without a subclinical, proxy diagnosis of ADHD (hypothesis 2). PA level was entered as the dependant variable (Y), ADHD symptomatology as the independent variable (X), and ADHD proxy diagnosis as the moderator (W) (see Fig 1).

To determine whether there was a significant difference in PA level based on motivation for exercise (hypothesis 3), a Kruskal-Wallis Test was conducted. Finally, a Spearman's rank correlation was performed to assess the relationship between TFE and ADHD, allowing examination of whether ADHD symptomatology differed based on main form of exercise

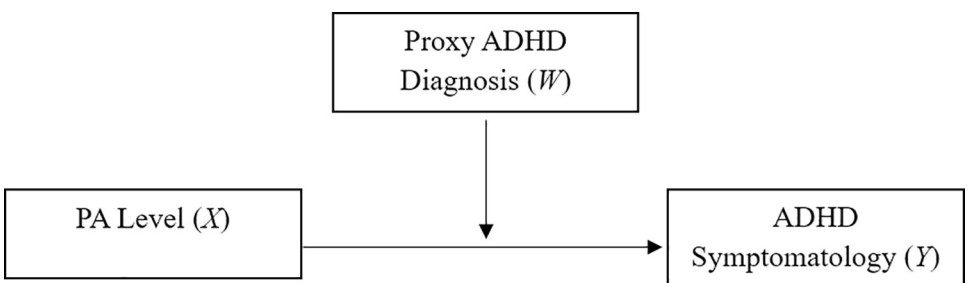

**Fig 1. Model for the moderation effect of ADHD proxy diagnosis between PA level and ADHD symptomatology.**

(hypothesis 4). A regression analysis was additionally performed to ensure that findings were not affected by PA level acting as a mediator between TFE and ADHD symptomatology (as PA level and TFE had a high level of correlation).

## 3. Results

### 3.1. Descriptive and demographic statistics

Table 1 shows the mean (interquartile range deviations) for the variables. Preliminary analyses of demographic variables, correlation coefficients, and a correlation matrix of scatterplots of main study variables can be found in the supplementary material (S3 Appendix, S2 Table and S1 Fig, respectively).

Table 2 shows the frequencies for Gender and Main motivation for exercise split by ADHD proxy diagnostic group. Visual depictions of bar charts can be found in supplementary material (S2 and S3 Figs).

Table 2 shows that within the proxy diagnostic group, there was a slightly higher ratio of females (76.92%), compared to the non-ADHD group (73.17%). However, a chi-square test found no significant difference in the prevalence of proxy diagnosis of ADHD between males and females, $\chi2(1) = 0.79$, $p = .376$, $\varphi = .054$ (three participants who selected 'other' or 'prefer not to say' were not included in the analysis).

**Table 1. Descriptive statistics for study variables.**

| | Sample/Group | | | | | | | | |
|---|---|---|---|---|---|---|---|---|---|
| | Total Sample (n = 268) | | | Non-ADHD Group (n = 164) | | | Proxy Diagnostic Group (n = 104) | | |
| Variables | Mean (SD) | Min | Max | Mean (SD) | Min | Max | Mean (SD) | Min | Max |
| Age | 25.88 (9.08) | 18 | 71 | 27.37 (10.34) | 18 | 71 | 23.54 (5.95) | 18 | 58 |
| ASRS-6 | 11.94 (4.08) | 2 | 24 | 9.49 (2.62) | 2 | 15 | 15.81 (2.75) | 11 | 24 |
| T-ADHD-S | 18.42 (9.54) | 0 | 46 | 14.54 (7.99) | 0 | 37 | 24.53 (8.57) | 8 | 46 |
| I-ADHD-S | 10.03 (5.19) | 0 | 22 | 7.81 (4.33) | 0 | 18 | 13.53 (4.48) | 5 | 22 |
| H-ADHD-S | 8.39 (5.53) | 0 | 25 | 6.73 (4.92) | 0 | 22 | 11 (5.45) | 0 | 25 |
| PA level | 3578.47 (3437.25) | 0 | 173739 | 3772.61 (3521.99) | 0 | 17379 | 3272.33 (3292.65) | 0 | 16458 |
| TNC | 0.66 (1.02) | 0 | 5 | 0.54 (0.91) | 0 | 4 | 0.87 (1.15) | 0 | 5 |
| TFE | 1.75 (1.42) | 0 | 7 | 1.84 (1.38) | 0 | 7 | 1.61 (1.48) | 0 | 7 |

Note: ASRS-6 = The Adult ADHD Self Report Scale score– 6; PA = Physical Activity; T-ADHD-S = Total ADHD Symptomatology; I-ADHD-S = Inattentive ADHD Symptomatology; H-ADHD-S = Hyperactive ADHD Symptomatology; TNC = Total Number of Comorbidities; TFE = Total Forms of Exercise.

**Table 2. Frequencies of gender and main motivation for exercise.**

| | Sample/Group | | |
| --- | --- | --- | --- |
| | Total Sample (n = 268, 100%) | Non-ADHD Group (n = 164, 61.19%) | Proxy-Diagnostic Group (n = 104, 38.81%) |
| Variables | n (% of total) | n (% of subgroup) | n (% of subgroup) |
| Gender | | | |
| Male | 65 (24.25) | 43 (26.22) | 22 (21.15) |
| Female | 200 (74.63) | 120 (73.17) | 80 (76.92) |
| Other | 2 (0.75) | 0 (0) | 2 (0.75) |
| Prefer not to say | 1 (0.37) | 1 (0.37) | 0 (0) |
| Main motivation for exercise | | | |
| Fitness and health | 149 (55.6) | 100 (60.98) | 49 (47.12) |
| To lose weight | 41 (15.3) | 24 (14.63) | 17 (16.35) |
| For social reasons | 11 (4.1) | 4 (2.44) | 7 (6.73) |
| Self defence | 1 (0.37) | 1 (0.61) | 0 (0) |
| For my appearance | 27 (10.45) | 16 (9.76) | 11 (10.58) |
| Do not exercise | 28 (10.45) | 12 (7.32) | 16 (15.38) |
| Other | 11 (4.1) | 7 (4.27) | 4 (3.85) |

## 3.2. Relationship between PA level and ADHD symptomatology

To investigate the relationship between PA level and ADHD symptomatology (hypothesis 1), a three-step hierarchical multiple regression was conducted with T-ADHD-S as the dependant variable, and age and TNC added as control variables (Table 3). Age was entered in Step one, TNC was entered in Step two, and PA level was entered in Step three. Prior to conducting a hierarchical multiple regression, the relevant assumptions of this statistical analysis were tested. Firstly, a sample size of 268 was considered sufficient for 3 independent variables to be included in the analysis. The assumption of singularity was met as the independent variables (Age, TNC and PA level) were not a combination of other independent variables. An examination of correlations revealed that no independent variables were highly correlated with one another, and collinearity statistics (i.e. tolerance and VIF) were all within acceptable limits. Thus, the assumption of multicollinearity was deemed to have been met. In addition, residual

**Table 3. Multiple hierarchical regression on total ADHD symptomatology.**

| Predictor Variables | R | $R^2 \Delta$ | p | B | 95% CI | SE | β | t |
| --- | --- | --- | --- | --- | --- | --- | --- | --- |
| **Step 1** | .14 | .02* | .023 | | | | | |
| Age | | | .023 | -.15 | [-.27, -.02] | .06* | -.14 | -2.29 |
| **Step 2** | .35 | .1** | < .001 | | | | | |
| Age | | | .034 | -.13 | [-.25, -.01] | .06* | -.12 | -2.13 |
| TNC | | | < .001 | 2.98 | [1.92, 4.04] | .54** | .32 | 5.52 |
| **Step 3** | .35 | >.01 | .536 | | | | | |
| Age | | | .036 | -.13 | [-.25, -.01] | .06* | -.12 | -2.11 |
| TNC | | | < .001 | 2.98 | [1.92, 4.04] | .54** | .32 | 5.51 |
| PA level | | | .536 | >-.01 | [>-.01, < .01] | < .01 | -.04 | -.62 |

*Note*. Statistical significance

*$p < .05$

**$p < .01$. B, represents unstandardised regression weights; CI represents confidence intervals for B; β, represents standardised regression weights. TNC = Total Number of Comorbidities; PA = Physical Activity.

and scatter plots indicated that the assumptions of normality, linearity and homoscedasticity were all met.

In Step one, age significantly contributed to the regression model, $F(1,266) = 5.22$, $p = .023$, accounting for approximately 1.9% of the variance in T-ADHD-S. When TNC was entered in Step two, the overall regression model remained significant, $F(2,265) = 18.15$, $p = < .001$), with TNC accounting for an additional 10.1% of the variance in T-ADHD-S. When PA level was entered in Step three, the overall regression model remained significant $F(3,264) = 12.2$, $p = < .001$), and accounted for 12.2% of the variance–but PA level only accounted for an additional 0.1% of variance and was not a significant unique predictor. In contrast, Age and TNC were both significant unique predictors, with the latter being the strongest overall predictor. Thus, these findings suggest that PA level and total ADHD symptomatology are not significantly associated.

To investigate relationships between PA and ADHD symptom subtypes, a series of Spearman's rho correlations were performed within the different subgroups (Table 4). Results

**Table 4. Descriptive statistics and correlation coefficients (Spearman's rho) for ADHD symptom and PA subtypes, split by proxy ADHD diagnosis and gender.**

| Variable | Median | IQR | 1 | 2 | 3 | 4 | 5 |
|---|---|---|---|---|---|---|---|
| Total Sample (n = 268) | | | | | | | |
| 1. T-ADHD-S | 17 | 13 | - | | | | |
| 2. I-ADHD-S | 10 | 8 | .88** | - | | | |
| 3. H-ADHD-S | 7 | 7 | .87** | .55** | - | | |
| 4. PA level | 2574.25 | 4031.25 | -.07 | -.14* | .03 | - | |
| 5. PA level (Leisure) | 594 | 1608.75 | -.13* | -.2** | -.03 | .69** | - |
| Non ADHD (n = 164) | | | | | | | |
| 1. T-ADHD-S | 14 | 9 | - | | | | |
| 2. I-ADHD-S | 8 | 7 | .82** | - | | | |
| 3. H-ADHD-S | 6 | 7 | .84** | .41** | - | | |
| 4. PA level | 2712 | 4211.25 | .01 | -.08 | .1 | - | |
| 5. PA level (Leisure) | 619 | 1864.13 | -.06 | -.17* | .09 | .72** | - |
| Proxy ADHD group (n = 104) | | | | | | | |
| 1. T-ADHD-S | 24 | 11.75 | - | | | | |
| 2. I-ADHD-S | 13 | 7 | .83* | - | | | |
| 3. H-ADHD-S | 11 | 8 | .86* | .45** | - | | |
| 4. PA level | 2388.75 | 3244.88 | -.11 | -.24* | .04 | - | |
| 5. PA level (Leisure) | 429 | 1358.63 | -.19 | -.23* | -.1 | .63** | - |
| Male (n = 65) | | | | | | | |
| 1. T-ADHD-S | 17 | 9 | - | | | | |
| 2. I-ADHD-S | 10 | 6.5 | .77* | - | | | |
| 3. H-ADHD-S | 8 | 6.5 | .8** | .29* | - | | |
| 4. PA level | 2760 | 4080 | .1 | -.05 | .24 | - | |
| 5. PA level (Leisure) | 396 | 1456.5 | -.02 | >.01 | >.01 | .71** | - |
| Female (n = 200) | | | | | | | |
| 1. T-ADHD-S | 17 | 15 | - | | | | |
| 2. I-ADHD-S | 10 | 8 | .9** | - | | | |
| 3. H-ADHD-S | 7 | 8 | .89** | .61** | - | | |
| 4. PA level | 2574.25 | 4016.25 | -.13 | -.18** | -.04 | - | |
| 5. PA level (Leisure) | 594 | 1642.5 | -.18* | -.27** | -.05 | .68** | - |

*Note.* Statistical significance

*$p < .05$

**$p < .01$. T-ADHD-S = Total ADHD Symptomatology; I-ADHD-S = Inattentive ADHD Symptomatology; H-ADHD-S = Hyperactive ADHD Symptomatology.

showed that although there was no significant relationship between PA and T-ADHD-S in the total sample (nor between PA and H-ADHD-S), there was a significant negative correlation between PA and I-ADHD-S, $r(266) = -.14$, $p = .02$, and a significant negative correlation between PA leisure and T-ADHD-S, $r(266) = -.13$, $p = .03$. This suggests that while there may not be a significant relationship between overall PA level and total ADHD symptomatology, significant negative relationships emerge when PA and ADHD symptomatology is split into sub-measures, such as PA leisure (representing more Physical Exercise) and inattentive ADHD symptoms.

Further, PA was not significantly correlated with any ADHD symptom types in the non-ADHD group, but there were significant negative correlations between PA and I-ADHD-S in the proxy diagnostic group $r(102) = -.24$, $p = .016$. Similarly, while PA was not significantly correlated with any ADHD symptom types in males, there was a significant negative correlation between PA and I-ADHD-S in females $r(198) = -.18$, $p = .01$). This suggests that that any significant negative correlations between PA and inattentive ADHD symptomatology might be specific to proxy ADHD individuals and females.

### 3.3. Effect of proxy ADHD diagnostic status on the relationship between PA level and ADHD symptomatology

The moderating role of proxy diagnosis of ADHD on the relationship between PA level and ADHD symptomatology (hypothesis 2) was assessed to determine if the strength of the correlation was different between proxy diagnostic ADHD participants and non-ADHD participants. A test of unconditional interaction was performed using PROCESS macro (with proxy ADHD diagnosis as moderator, PA level as predictor, and T-ADHD-S as the outcome), with the change in $R^2$ due to interaction of predictor and moderator (Table 5). Altogether, 26.12% of the variability in T-ADHD-S was predicted by the variables, $R^2 = .26$, $F(3,264) = 31.11$, $p < .001$. The interaction (shown in Fig 2) was not significant, as the impact of PA level on T-ADHD-S in non-ADHD participants was not significantly different from ADHD proxy diagnostic participants ($b = 0$, 95% C.I. [$>-.01$, $< .01$], $t = .242$, $p = .809$). Table 6 presents the conditional effect of PA at the two levels of Proxy ADHD diagnosis. This suggests that the relationship between PA and ADHD symptomatology is not significantly different based on ADHD diagnostic status.

### 3.4. Effect of motivation for exercise on PA level

To determine whether there was a significant difference in PA level based on the main form of motivation for exercise (e.g., "For social reasons", "For my appearance"), a Kruskal-Wallis Test was conducted (hypothesis 3). No significant differences, $H(5) = 5.10$, $p = .404$) were found across the five categories of participants. Therefore, this suggests that PA level does not differ based on an individual's main form of motivation for exercise.

**Table 5. Moderated regression analysis predicting ADHD symptomatology.**

| Predictor | B | SE | t | p | 95% CI |
|---|---|---|---|---|---|
| Constant | 14.54 | 0.65 | 22.54 | < .001 | [13.27, 15.81] |
| PA Level | < .01 | < .01 | 0.06 | .949 | [>-.01, < .01] |
| PAD | 9.97 | 1.04 | 9.61 | < .001 | [7.92, 12.01] |
| PA Level*PAD | >-.01 | < .01 | -0.24 | .809 | [>-.01, < .01] |

Note: B, represents unstandardised regression weights; CI represents confidence intervals for B. PA = Physical Activity; PAD = Proxy ADHD Diagnosis.

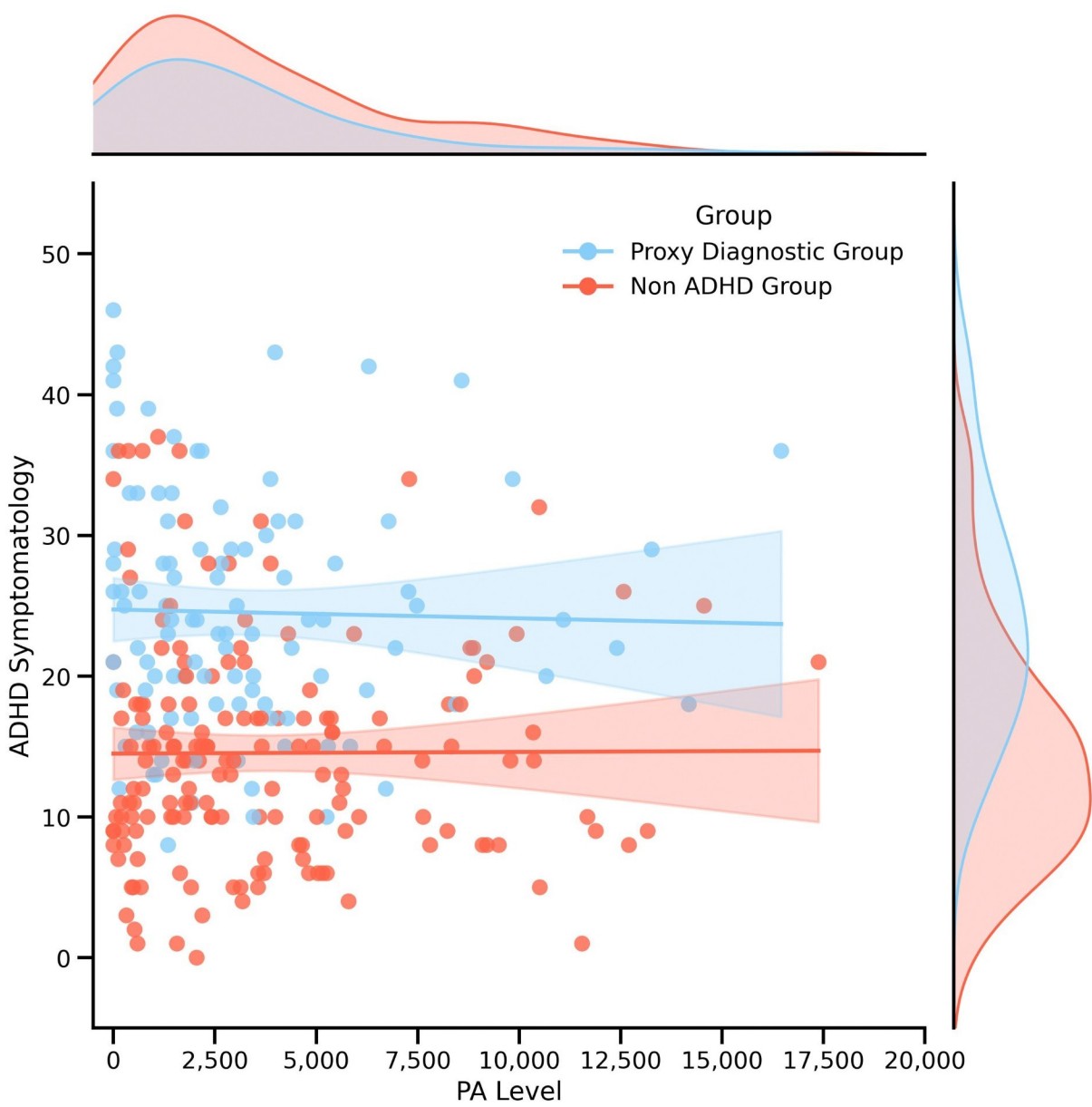

**Fig 2. Interaction between total ADHD symptomatology and PA level based on proxy ADHD diagnosis.**

**Table 6. Conditional effects of PA level across proxy ADHD diagnosis.**

| PAD | Effect | SE | t | p | 95% CI |
|---|---|---|---|---|---|
| NonADHD | < .01 | < .01 | 0.06 | .949 | [>-.01, < .01] |
| ProxyADHD | >-0.01 | < .01 | -0.25 | .8 | [>-.01, < .01] |

Note: CI represents confidence intervals for conditional effects. PA = Physical Activity

Table 7. Multiple hierarchical regression on total ADHD symptomatology.

| Predictor Variables | R | $R^2\Delta$ | p | B | 95% CI | SE | β | t |
|---|---|---|---|---|---|---|---|---|
| Step 1 | .04 | >.01 | .498 | | | | | |
| PA level | | | .498 | >-.01 | [>-.01, <.01] | <.01 | -.04 | -.68 |
| Step 2 | .11 | .01 | .115 | | | | | |
| PA level | | | .899 | <.01 | [>-.01, <.01] | <.01 | .01 | .13 |
| TFE | | | .115 | -.73 | [-1.64, .18] | .46 | -.11 | -1.58 |

Note: B, represents unstandardised regression weights; CI represents confidence intervals for B; β, represents standardised regression weights. PA = Physical Activity; TFE = Total Forms of Exercise.

### 3.5. Relationship between forms of exercise and ADHD symptomatology

Next, In relation to whether ADHD symptomatology differed based on main form of exercise (hypothesis 4), a Spearman's rank correlation confirmed that there was no significant correlation between TFE and T-ADHD-S, $r_s(266) = -.114$, $p = .063$. However, to ensure that the findings were not affected by PA level acting as a mediator between TFE and T-ADHD-S (as PA level and TFE had a high level of correlation), PA level was also controlled for in a regression analysis. Specifically, a two-stage hierarchical multiple regression was conducted, with T-ADHD-S as the dependent variable (Table 7). PA level was entered in Step one, and TFE entered in Step two. In Step one, PA did not contribute significantly to the regression model, $F(1,266) = 0.48$, $p = .498$, and accounted for only 0.2% of the variation in T-ADHD-S. When TFE was entered in Step two, it accounted of for an additional 0.9% of the variance in T-ADHD-S, and the overall model was not significant, $F(1,265) = 2.50$, $p = .115$). In total, the final model accounted for 1.1% of the variance in T-ADHD-S, which was not significant. As such, this suggests that there is no significant relationship between total ADHD symptomatology and the number of forms of exercise engaged in (after controlling for PA level).

## 4. Discussion

Building on initial research primarily conducted with children [27–31], the overarching aim of this study was to investigate the relationship between levels of PA and ADHD symptoms in adults, while exploring the effects of potential moderating variables. Addressing some of the limitations of prior research in this area, a further aim was to investigate the relation between PA and ADHD symptoms among participants who may not have a formal diagnosis, but who may reach psychometrically-defined subclinical (i.e., proxy) thresholds for ADHD. In contrast to prior research [27–31], there was no significant correlation between ADHD symptom severity or PA level. There was also no significant difference in the strength of correlation between ADHD symptom severity or PA level based on proxy ADHD diagnosis. There was however a significant negative correlation between inattentive ADHD symptom severity and PA level, and between ADHD symptom severity and PA level in the leisure domain. Extending prior research [29, 61, 62] no significant difference in PA level based on motivation for exercise was found, and there was also no significant difference in ADHD symptomatology based on main form of exercise. This means that hypothesis 1 was partially supported; hypothesis 2 was supported, and hypotheses 3 and 4 were not supported. The main findings of the study are discussed further below.

Based on prior literature [27–31], it was hypothesised that there would be a significant negative correlation between PA level and ADHD symptomatology (hypothesis 1). In contrast to this, no significant correlation between PA level and total ADHD symptomatology was found

in the sample as a whole. This contrasts with previous research by Abramovitch et al. [43] who found that individuals engaging in high levels of PA reported significantly fewer ADHD symptoms (i.e., behavioural impulsivity and worrisome and intrusive thoughts). When spilt by ADHD subtype in this study though, PA was found to have a significant negative correlation with inattentive ADHD symptomatology, and a positive (albeit not significant) relationship with hyperactive ADHD symptomatology. The difference between the strength of these two correlations was found to be statistically significant also [73], potentially suggesting that PA may only have a significant effect on reducing inattentive ADHD symptoms, and not hyperactive ones. The other possibility though is that individuals with predominantly hyperactive ADHD symptoms are more likely to engage in PA than individuals with predominantly inattentive ADHD symptoms. This would align with prior research finding that inattentive symptoms in childhood predicted reduced PA in adolescence, whilst the opposite was true for hyperactive symptoms [74], and that individuals with hyperactive presentations may engage with PA as a way of coping and self-treating their symptoms. However, these possible explanations are not mutually exclusive (and both could potentially be contributing to the observed findings) and would need to be investigated further via alternative experimental designs before drawing more definitive conclusions.

Furthermore, greater understanding can also come from breaking down the PA measurement into more specific subcategories, with PA from the leisure domain showing a significant negative correlation with general ADHD symptoms and inattentive symptoms, but a positive (non-significant) correlation with hyperactive ADHD symptoms. This could suggest that any positive effects of PA on ADHD symptoms in adults may be reserved to physical exercise, rather than other forms of physical activity. This view is supported by existing research that has found increased benefits of voluntary exercise (which would likely be the case in the leisure domain), compared to involuntary and forced exercise (which would be more similar to the other domains) [75–77]. This finding could also support the earlier suggestion that individuals with predominantly hyperactive ADHD symptoms are more likely to engage in PA than individuals with predominantly inattentive ADHD symptoms. Either way though, these nuanced findings highlight the importance of distinguishing between physical activity versus physical exercise when exploring the relationship between PA level and ADHD symptomatology.

However, our findings lend support for the second hypothesis, that there would be no significant difference between the strength of relationship between PA level and ADHD symptomatology in participants with and without a proxy ADHD diagnosis. Again, the findings initially appear to contradict existing research reporting that the rate of improvement of PA on ADHD symptomatology is slightly greater in ADHD subjects [41, 57, 58]. For example, Pontifex et al. [57] concluded that only children with diagnosed ADHD exhibited exercise-induced facilitations in action monitoring processes, and regulatory adjustments in behaviour, after exercise. One explanation for the apparent difference in findings, is that previous research only observed a significant *acute* effect of PA on ADHD. In contrast, the measure of ADHD symptomatology in this study would likely only reflect a *chronic* effect of PA. If the differences in effect size between ADHD and non-ADHD subjects is due to acute and not chronic term effects, then this could not be reflected in a correlation study such as this. Additionally, Wigal et al. [59] found that exercise caused a significantly greater increase in dopamine, norepinephrine and epinephrine for children without than with ADHD. It is also important to note, that when looking at hyperactive or inattentive ADHD symptoms specifically when split by proxy diagnosis, the only significant correlation was a negative one between inattentive ADHD symptoms and PA level in ADHD proxy diagnostic participants. This could suggest that any positive relationships between, and potential effects of PA on ADHD symptomatology, were not only exclusive to inattentive ADHD symptoms, but also only reached a significant level for

subjects with a proxy, subclinical level of ADHD. This could also partially explain initial differences in findings highlighted earlier with Abramovitch et al. [43], as they only sampled individuals with ADHD. However, owing to the correlational design of the current study, these findings could also suggest that only subjects with a proxy, subclinical diagnosis of ADHD as well as predominantly inattentive symptoms, are significantly less likely to engage in PA than other subjects (subjects without a proxy, subclinical diagnosis of ADHD; and subjects with a proxy, subclinical diagnosis but with predominantly hyperactive symptoms). However, a groupwise correlation analysis found that there was no significant difference in the strength of correlation between PA level and inattentive ADHD symptoms when split by proxy ADHD diagnosis (despite one correlation being significant, and one being non-significant), suggesting that not too much weight should be given to this possibility.

It was also hypothesised that there would be a significant difference in PA level based on motivation for exercise (hypothesis 3), which was not supported by the current data. While most research into motivation and PA has been focused on levels of motivation (and therefore is not applicable to the data in this study), the findings of this study, in regard to motivation type, contradict the limited amount of relevant research. No significant difference was found in PA level based on motivation type, which is partially supported by previous research from Cash et al. [78] who found that there was no significant correlation between exercise frequency and *most* reasons for exercise. However, direct comparison is difficult as their sample was exclusively female and they also used different measurement tools (e.g., motivation was measured using the Reasons for Exercise Inventory and PA was measured by weekly exercise frequency). Additionally, our findings conflict with research from Duncan et al. [62], who found that greater frequency, duration, and intensity of exercise were correlated with more autonomous than controlling regulations. Reasons for this discrepancy are likely due to differences in the measurement of motivation: while the current study based different motivation types around potential different reasons, Duncan et al. [62] based it around self-determination theory (e.g., intrinsic; identified; introjected; and external). This could suggest that while the precise reasons for motivation may not be important for improving PA engagement (and any potential PA based intervention), their source (internal versus external) may be. This could be particularly relevant to adults with ADHD as the extrinsic motivations for children to engage in interventions (e.g., compelled to by parents/school) would be less present for adults who would rely more on intrinsic motivations [79].

Unfortunately, it was not possible to fully explore our hypothesis that there would be a significant difference on ADHD symptomatology based on main form of exercise (hypothesis 4), as it was not possible to effectively group different forms of exercise. Therefore, the total number of different forms of exercise performed by an individual were instead investigated. Previous literature has suggested that more combined based PA interventions (involving both aerobic and strength training) can lead to more positive benefits than singular based interventions [29, 61]. Our findings seemed to conflict with this as there was no significant correlation between total forms of exercise performed by a participant and ADHD symptomatology. However, like the main relationship between PA level and ADHD symptomatology, when ADHD symptomatology was split by type, it was found that hyperactive ADHD symptoms had a non-significant positive correlation with total forms of exercise, whereas inattentive symptoms had a significant negative correlation. Hypothetically, this could suggest that any positive effects of combining different exercise types are only found in inattentive ADHD symptoms/subtypes; and/or individuals with predominantly inattentive ADHD symptoms are less likely to try out different forms of exercise. However, note that the exact direction and nature of any such effects would have to be determined via additional work with an alternative study design.

Another unexpected finding was the lack of any significant difference in proxy diagnostic rates of ADHD when split by gender. As already outlined, previous research has focused largely on males due to higher diagnosis rates, potentially overlooking a large sample of females with ADHD due to underdiagnosis. The findings of the study potentially support the claim of female underdiagnosis in ADHD [55] and suggest that the prevalence rate is much more even across males and females. Of particular interest was the finding that when the data was split by gender, the negative correlation between inattentive ADHD symptoms and PA level was not significant in males but reached a high level of significance in females ($p = .01$).

One possible implication of our findings to policy and practice, is that efforts to incorporate exercise into treatment regimens for adult ADHD may benefit from being focused towards individuals with the predominantly inattentive presentation of ADHD than other presentations (combined or hyperactive), either because: the positive effects of exercise are most significant in inattentive ADHD symptoms; and/or individuals with predominantly inattentive ADHD are less likely to already be engaged in exercise than those with other presentations. However, owing to the correlational nature of our design, it should be acknowledged that our findings could also suggest instead that individuals with hyperactive ADHD may be more open and likely to engage with exercise-based interventions. This could then imply the opposite: that exercise treatments could have greater effect when focused on individuals with hyperactive than inattentive symptomatology. Thus, a dual approach may be beneficial: encouraging exercise among individuals with a predominantly inattentive subtype, whilst leveraging the tendency for those with hyperactive presentations to perhaps already engage in PA.

As there was no significant difference between motivation types on PA level, this also suggests that motivation might not be a variable of importance when designing potential exercise interventions. Instead, it suggests that if an individual can be motivated in some way, and this motivation is primarily internal rather than external, this should lead to increased commitment with any interventions [62]. Therefore, rather than interventions focusing their promotion of exercise for a particular reason (e.g., focusing on specifically the health benefits, or social aspects, or appearance etc), which might not be important or relevant to individuals (and act as external regulations), professionals may benefit from adopting an individualistic approach: taking time with individuals to find out what potential motivations are the most important to them specifically (acting as internal regulations), and focusing on those to promote exercise. That is, interventions should be personalised, identifying and fostering the most relevant motivations for each individual. In turn, this could enhance adherence to, and potentially the effectiveness of, PA-based interventions. It is important to note though that this study was not able to differentiate between the different forms of exercise and consequently, it is unclear if any particular form of exercise should be promoted over others.

## 4.1. Limitations and future directions

Our study has a number of limitations that should be taken into account when considering its implications for practice (i.e., intervention design) and suggestions for further research. Data collection occurred between December 2021 and March 2022, at a time when COVID-19 lockdown conditions were largely in place in the UK. This could have affected participants answers on the IPAQ-L, as they might not have been able to engage in as much PA as normal. The sample was also biased towards females, students, and younger adults. This may have contributed towards the main measures of PA and ADHD symptomatology not being normally distributed as students may engage in less PA than the general population, thus leading to the negative skew in the IPAQ-L. Therefore, given the composition of the sample and study context, our findings may not be generalisable to a more diverse population. A lack of specificity in some

measurement types also limited the amount of analysis possible. Thus, future studies would benefit from utilising more detailed and specific measures (i.e., types of exercise, motivation) where practicable and feasible, to allow a more nuanced analysis and understanding of their potential effects.

Additionally, future research could profitably focus on investigating potential cause and effect relationships between PA and ADHD symptomatology. Longitudinal research could focus on how likely adult individuals with ADHD are to undertake (and continue to engage with) PA and/or how long-term PA effects different ADHD symptomatology. The effectiveness of PA on different ADHD symptoms would need to be considered alongside how likely individuals with different ADHD presentations are to engage with PA (during the development of any potential PA treatment) to ensure that it would have the greatest efficacy.

To enable further evaluation of the potential effectiveness of PA on ADHD symptomatology, and/or the likelihood of those with ADHD fruitfully engaging with PA, future work may also wish to explore different types of ADHD symptoms and presentations, and to also differentiate between inattentive and hyperactive subtypes, rather than treat ADHD as a monolith category [27–31]. In turn, additional insights gained could help improve the efficacy and potential cost-effectiveness of any interventions also (e.g., by allowing for greater determination on who would benefit the most from/be more likely to engage with any treatments).

Similarly, the findings highlight the importance of further addressing ADHD in females in this area. Using subclinical/proxy diagnostic measures might allow for a much higher rate of female participation, as well as supporting future research to more easily reach sufficient sample sizes for robust analysis. Secondly, the findings suggest that the correlation between PA and ADHD symptomatology is different between genders. Whether this is due to the effect of PA being different in females, or because ADHD in females makes them less inclined to perform PA than males, is currently unclear and warrants further investigation. As such, future research should try to distinguish between demographic factors such as gender and ADHD presentation type, as failing to do so may lead to overlooking potentially significant differences among individuals with ADHD. Indeed, considering the nature of ADHD and its potential interactions with various external factors, it is important that future research more broadly considers potential confounding variables throughout the research process. For instance, PA preferences and energy levels may vary across age, comorbidities may interact with ADHD symptoms and affect individuals' motivation or ability to engage in PA, and lockdown circumstances may result in decreased PA levels.

Additionally future research could focus on developing evidence-based standardisations for different aspects measured in this study, to allow for greater analytical specificity. Research into the effects and relationships of different types of motivation and forms of exercise would be greatly enhanced by more clear-cut categorisations, also allowing for greater meta-analysis and comparison between studies.

## 4.2. Conclusion

To conclude, the findings of this study suggest that the relationship between PA and ADHD symptomatology is complex, and may need to be broken down into smaller, more specific factors to be properly understood. Specifically, while there was no significant relationship between PA level and total ADHD symptomatology, further analysis suggests that significant relationships may exist between specific subtypes of PA (such as physical exercise) and ADHD (such as inattentive ADHD symptoms), as well as within specific subgroups (such as females and individuals with ADHD). This underscores the need for additional detailed investigations to further elucidate the relationship between PA level and ADHD symptomatology in adults,

and to enable evaluation of the potential effectiveness of PA as a potential adjunct therapeutic option.

## Supporting information

**S1 Checklist. STROBE statement.**
(DOCX)

**S1 Appendix. Psychometric properties of measures used.**
(DOCX)

**S2 Appendix. Procedure and questionnaire.**
(DOCX)

**S3 Appendix. Preliminary analysis of results.**
(DOCX)

**S1 Fig. Correlations matrix of scatterplots and histograms for ADHD symptomatology, PA level; Mood; TFE; TNC and age.**
(TIF)

**S2 Fig. Frequencies of gender across proxy diagnostic groups.**
(TIF)

**S3 Fig. Frequencies of main motivation for exercise across proxy diagnostic groups.**
(TIF)

**S1 Table. Demographic information of the study sample.**
(DOCX)

**S2 Table. Descriptive statistics and correlation coefficients (Spearman's rho) for study variables (n = 268).**
(DOCX)

## Author Contributions

**Conceptualization:** Rory Tucker, Claire Williams, Phil Reed.

**Data curation:** Rory Tucker.

**Formal analysis:** Rory Tucker, Claire Williams, Phil Reed.

**Investigation:** Rory Tucker.

**Methodology:** Rory Tucker, Claire Williams, Phil Reed.

**Project administration:** Rory Tucker, Claire Williams, Phil Reed.

**Supervision:** Claire Williams, Phil Reed.

**Writing – original draft:** Rory Tucker.

**Writing – review & editing:** Rory Tucker, Claire Williams, Phil Reed.

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
