## [Decision Letter · Decision Letter 0]

31 Jul 2023

PONE-D-23-16228Association of exercise and ADHD symptoms: Analysis within an adult general population samplePLOS ONE

Dear Dr. Tucker,

Thank you for submitting your manuscript to PLOS ONE. After careful consideration, we feel that it has merit but does not fully meet PLOS ONE’s publication criteria as it currently stands. Therefore, we invite you to submit a revised version of the manuscript that addresses the points raised during the review process. Please submit your revised manuscript by Sep 14 2023 11:59PM. If you will need more time than this to complete your revisions, please reply to this message or contact the journal office at plosone@plos.org. Please include the following items when submitting your revised manuscript:A rebuttal letter that responds to each point raised by the academic editor and reviewer(s). You should upload this letter as a separate file labeled 'Response to Reviewers'.A marked-up copy of your manuscript that highlights changes made to the original version. You should upload this as a separate file labeled 'Revised Manuscript with Track Changes'.An unmarked version of your revised paper without tracked changes. You should upload this as a separate file labeled 'Manuscript'.

We look forward to receiving your revised manuscript.

Kind regards,

Giuseppe Marano

Academic Editor

PLOS ONE

Reviewers' comments:

Reviewer's Responses to Questions

**Comments to the Author**

1. Is the manuscript technically sound, and do the data support the conclusions?

Reviewer #1: Partly

Reviewer #2: Partly

Reviewer #3: Yes

2. Has the statistical analysis been performed appropriately and rigorously? 

Reviewer #1: Yes

Reviewer #2: I Don't Know

Reviewer #3: Yes

3. Have the authors made all data underlying the findings in their manuscript fully available?

Reviewer #1: Yes

Reviewer #2: Yes

Reviewer #3: Yes

4. Is the manuscript presented in an intelligible fashion and written in standard English?

Reviewer #1: Yes

Reviewer #2: No

Reviewer #3: Yes

5. Review Comments to the Author

Reviewer #1: The article entitled “ Association of exercise and ADHD symptoms: Analysis within an adult general

population sample” explores an important topic regarding ADHD symptomatology which can lead to new approaches for ADHD management. However there are main issues which could be considered.

- The main conceptualization of the topic and its elaborations needs a major revision. The study is a descriptive cross sectional one which describes the relationship between physical activity and ADHD symptoms. However, the most parts of the introduction and discussion is devoted to the possible positive effects of physical activity “interventions” on ADHD symptoms. Such conclusion or attribution can not be obtained from the manuscript.

- Table 1 should be inserted in Results section. Abbreviations should be fully explained under the tables.

- It is mentioned that “The ADHD Rating Scale – IV with Adult prompts” has 9 subscales but only two of them was explained

- PHQ-5 and The Big Five Inventory – 10 has been utilized in methodology but there are no data regarding these two questionnaires in the results.

- Presentation of data in the results should be reordered. It should be initiated with demographic data following descriptive data and results of correlational analysis should be presented afterwards. Starting the result section with a table (Table 2) is not preferred. Moreover, table 2 is not clearly explained.

- Line 477-502 does not have relevancy with the main purpose of the manuscript

- Line 526-536: similar to what has been said in first comment, the conclusion you have mentioned cannot be speculated from your findings.

Reviewer #2: Please see attached document. I have answered 'I don't know' to the question "Has the statistical analysis been performed appropriately and rigorously?" as there is some missing information in the MS around analyses at this stage.

Reviewer #3: Introduction

You should add more references/studies on the psychiatrics/clinical aspects. You mentioned very well how ADHD can affect cost, but need be more specified how aggressive can be this disorder, often is underestimated compared to other psychiatric disorders. Add also something about the comorbidities and suicide risk.

Row 29-30: In contrast to research with children, please reformulate that.

Row 35: non specific variables/ measures are used. Give an example.

Row 56-57: sum up cost, you named it two times.

Row 82: improve cognitive function, name the most important.

Row 88: cognitive deficit, name the most important.

Row 145: a further aim was. I think you need a future form.

Discussion

Row 537 - 539 : reformulate the sentence.

Row 595- 599 : reformulate the sentence, maybe create two different sentences.

6. PLOS authors have the option to publish the peer review history of their article (what does this mean?). If published, this will include your full peer review and any attached files.

Reviewer #1: **Yes: **Mahtab Motamed

Reviewer #2: No

Reviewer #3: **Yes: **Maurizio Cundari

---

## [Author Response · Author response to Decision Letter 0]

25 Jan 2024

Dear Reviewers,

Thank you very much for your helpful and detailed feedback. We have responded to each of your suggestions and queries in the the document "Responce to Reviewers". We hope you find that the manuscript has been effectively revised and improved in responce to your comments.

---

## [Decision Letter · Decision Letter 1]

29 Apr 2024

PONE-D-23-16228R1Association of exercise and ADHD symptoms: Analysis within an adult general population samplePLOS ONE

Dear Dr. Tucker,

Thank you for submitting your manuscript to PLOS ONE. After careful consideration, we feel that it has merit but does not fully meet PLOS ONE’s publication criteria as it currently stands. Therefore, we invite you to submit a revised version of the manuscript that addresses the points raised during the review process.

We look forward to receiving your revised manuscript.

Kind regards,

Giuseppe Marano

Academic Editor

PLOS ONE

Reviewers' comments:

Reviewer's Responses to Questions

**Comments to the Author**

1. If the authors have adequately addressed your comments raised in a previous round of review and you feel that this manuscript is now acceptable for publication, you may indicate that here to bypass the “Comments to the Author” section, enter your conflict of interest statement in the “Confidential to Editor” section, and submit your "Accept" recommendation.

Reviewer #2: All comments have been addressed

Reviewer #3: All comments have been addressed

2. Is the manuscript technically sound, and do the data support the conclusions?

Reviewer #2: Yes

Reviewer #3: Yes

3. Has the statistical analysis been performed appropriately and rigorously? 

Reviewer #2: Yes

Reviewer #3: I Don't Know

4. Have the authors made all data underlying the findings in their manuscript fully available?

Reviewer #2: Yes

Reviewer #3: Yes

5. Is the manuscript presented in an intelligible fashion and written in standard English?

Reviewer #2: Yes

Reviewer #3: Yes

6. Review Comments to the Author

Reviewer #2: Thank you for addressing my extensive comments and concerns. Well done on producing a strong piece of work.

Reviewer #3: 55-57: "along with poor attention and memory, to the point that it interferes with development or functioning [1]."

Here I would like you to be more specific on cognitive deficits, is not only memory and attention involved in this disorder. Furthermore, not all memory domains is compromised but some memory aspects/functions.

Executive functions are also very important and nevertheless social cognition which is very little named for this disorder. Add some part of cognition and few references.

The other comments were changed correctly.

The other reviewers asked also question that I was wondering. I think is enough you reply to them and I follow the conversation.

I would like to see the final manuscript with all the changes and read it again.

59: please spell NHS.

89-96: try to sum it up.

110: symptomology (the point is missing) .

238: average hours slept; and fitness app use. Add References

306: table 1

Group

Total (n= 268) Non-ADHD (n=164) Proxy Diagnostic (n=104) - between group and next raw there is no space. Visually do not look very clear maybe write GROUP in capitals or add a space or a line.

- Same comments for table 2.

428-436 and 439: in the discussion, add references in the text of previous similar findings.

610-613: reformulate. and also clarify the role of confounders.

7. PLOS authors have the option to publish the peer review history of their article (what does this mean?). If published, this will include your full peer review and any attached files.

Reviewer #2: No

Reviewer #3: **Yes: **Maurizio Cundari

---

## [Author Response · Author response to Decision Letter 1]

30 Apr 2024

Dear Reviewers,

Thank you once again for your helpful feedback. We have responded to

your suggestions and queries in the the document "Responce to Reviewers".

We hope you find that the manuscript has been effectively revised and improved in

responce to your comments.

---

## [Decision Letter · Decision Letter 2]

2 Jul 2024

PONE-D-23-16228R2Association of exercise and ADHD symptoms: Analysis within an adult general population samplePLOS ONE

Dear Dr. Tucker,

Thank you for submitting your manuscript to PLOS ONE. After careful consideration, we feel that it has merit but does not fully meet PLOS ONE’s publication criteria as it currently stands. Therefore, we invite you to submit a revised version of the manuscript that addresses the points raised during the review process. Which changes you require for acceptance versus which changes you recommend

We look forward to receiving your revised manuscript.

Kind regards,

Giuseppe Marano

Academic Editor

PLOS ONE

Journal Requirements:

Reviewers' comments:

Reviewer's Responses to Questions

**Comments to the Author**

1. If the authors have adequately addressed your comments raised in a previous round of review and you feel that this manuscript is now acceptable for publication, you may indicate that here to bypass the “Comments to the Author” section, enter your conflict of interest statement in the “Confidential to Editor” section, and submit your "Accept" recommendation.

Reviewer #4: All comments have been addressed

Reviewer #5: All comments have been addressed

Reviewer #6: All comments have been addressed

2. Is the manuscript technically sound, and do the data support the conclusions?

Reviewer #4: Yes

Reviewer #5: Yes

Reviewer #6: Yes

3. Has the statistical analysis been performed appropriately and rigorously? 

Reviewer #4: Yes

Reviewer #5: Yes

Reviewer #6: Yes

4. Have the authors made all data underlying the findings in their manuscript fully available?

Reviewer #4: Yes

Reviewer #5: Yes

Reviewer #6: Yes

5. Is the manuscript presented in an intelligible fashion and written in standard English?

Reviewer #4: Yes

Reviewer #5: Yes

Reviewer #6: Yes

6. Review Comments to the Author

Reviewer #4: Based on previous reviewers ' recommendations, the article has undergone considerable editing and restructuring of the main concepts and methodology. For future studies/articles, it would behove the authors to carry out more thorough preparation and clarify conceptual matters before submitting for publication.

Reviewer #5: I want to congratulate the authors for the well-written work on an important topic that might influence the management of ADHD. I have a few comments that I hope will improve the paper.

1. I would suggest including a distinct subheading to present the study's conclusion. Perhaps, lines 617-629 could be included in the CONCLUSION section.

2. Furthermore, references lack complete data, such as DOIs. Providing this information is crucial to facilitating readers' access to the cited documents. It is essential to ensure that all references, including DOIs, are complete to enhance the comprehensiveness and accessibility of the bibliography.

Reviewer #6: (No Response)

7. PLOS authors have the option to publish the peer review history of their article (what does this mean?). If published, this will include your full peer review and any attached files.

Reviewer #4: No

Reviewer #5: No

Reviewer #6: No

---

## [Author Response · Author response to Decision Letter 2]

8 Jul 2024

Dear Reviewers,

Thank you very much for your helpful and detailed feedback. We have responded to each of your suggestions and queries in the the document "Responce to Reviewers". We hope you find that the manuscript has been effectively revised and improved in responce to your comments.

---

## [Decision Letter · Decision Letter 3]

12 Nov 2024

Association of exercise and ADHD symptoms: Analysis within an adult general population sample

PONE-D-23-16228R3

Dear Dr. Tucker,

We’re pleased to inform you that your manuscript has been judged scientifically suitable for publication and will be formally accepted for publication once it meets all outstanding technical requirements.

Kind regards,

Giuseppe Marano

Academic Editor

PLOS ONE

Additional Editor Comments (optional):

Reviewers' comments:

Reviewer's Responses to Questions

**Comments to the Author**

1. If the authors have adequately addressed your comments raised in a previous round of review and you feel that this manuscript is now acceptable for publication, you may indicate that here to bypass the “Comments to the Author” section, enter your conflict of interest statement in the “Confidential to Editor” section, and submit your "Accept" recommendation.

Reviewer #5: All comments have been addressed

2. Is the manuscript technically sound, and do the data support the conclusions?

Reviewer #5: Yes

3. Has the statistical analysis been performed appropriately and rigorously? 

Reviewer #5: Yes

4. Have the authors made all data underlying the findings in their manuscript fully available?

Reviewer #5: Yes

5. Is the manuscript presented in an intelligible fashion and written in standard English?

Reviewer #5: Yes

6. Review Comments to the Author

Reviewer #5: The authors have responded to comments from earlier reviews. I do not have further comments on the paper.

7. PLOS authors have the option to publish the peer review history of their article (what does this mean?). If published, this will include your full peer review and any attached files.

Reviewer #5: No

---

## [Editor Report · Acceptance letter]

18 Nov 2024

PONE-D-23-16228R3 

PLOS ONE

Dear Dr. Tucker, 

I'm pleased to inform you that your manuscript has been deemed suitable for publication in PLOS ONE. Congratulations! Your manuscript is now being handed over to our production team.

Kind regards, 

on behalf of

Dr. Giuseppe Marano 

Academic Editor

PLOS ONE